



# Measurement report: Lessons learned from the comparison and combination of fine carbonaceous aerosol source apportionment at two locations in the city of Strasbourg, France

Hasna Chebaicheb[1,2,3], Mélodie Chatain[4], Olivier Favez[2,3], Joel F. de Brito[1], Vincent Crenn[5], Tanguy Amodeo[2,3], Mohamed Gherras[2], Emmanuel Jantzem[4], Caroline Marchand[2,3], Véronique Riffault[1,3]

[1]IMT Nord Europe, Institut Mines-Télécom, Université de Lille, Centre for Energy and Environment, 59000, Lille, France
[2]Institut National de l'environnement Industriel et des Risques (INERIS), 60550 Verneuil-en-Halatte, France
[3]Laboratoire Central de Surveillance de la Qualité de l'Air (LCSQA), 60550 Verneuil-en-Halatte, France
[4]Atmo Grand Est, 67300 Schiltigheim, France
[5]ADDAIR, F-78530 Buc, France

*Correspondence to*: Hasna Chebaicheb (hasna.chebaicheb@ineris.fr); Mélodie Chatain (melodie.chatain@atmo-grandest.eu)

**Abstract.** Source apportionment analyses of carbonaceous aerosol were conducted at two neighboring urban sites in Strasbourg, France, during the winter of 2019/2020 using ACSMs (Aerosol Chemical Speciation Monitors; for non-refractory submicron aerosols), aethalometers (AE33; for equivalent Black Carbon - eBC) and filter-based offline chemical speciation. Positive Matrix Factorization (PMF) was applied to organic aerosols (OA) following two strategies: i) analyzing each site individually, ii) combining both sites into a single dataset. Both methods resolved five OA factors: hydrocarbon-like (HOA), biomass burning (BBOA), cooking-like (COA-like), oxygenated (OOA), and an amine-related OA (58-OA) factor. The latter factor, accounting for ~4% of the total OA mass at each site, showed clear diel profiles and a distinct origin marked by specific wind directions, suggesting a unique local source, potentially linked to industrial emissions. The present study also highlights the challenge of attributing a cooking-only origin to the COA-like factor, which exhibited a diel cycle similar to biomass burning OA at the background site. The combined PMF analysis improved the apportionment of cooking emissions at nighttime, especially for the traffic site, compared to individual PMF analyses, but it did not enhance the other OA factors due to instrumental specificities (i.e., different fragmentation patterns) leading to differences in OA mass spectra between the two instruments. Overall, this study argues for careful inspection of instrumental peculiarities in ACSM and AE33 data treatment and provides hints to benefit from their use at various locations at the city scale. It also allows comparison between different types of PMF analyses, showing that combined PMF may not be appropriate for improving the consistency of OA factors in some cases such as the one presented here.

**Keywords.** Urban pollution, PMF, organic aerosols, Amine-related OA



## 1 Introduction

Air pollution influences climate change and induces adverse effects on human health, increasing disease and mortality rates (EEA, 2022). In particular, particulate matter such as those with an aerodynamic diameter smaller than 1 µm ($PM_1$) are inhaled and reach the deeper respiratory system, leading to a range of health problems including respiratory and cardiovascular disorders, disruptions in reproductive and central nervous functionalities, as well as the development of cancer (WHO, 2022; Duarte et al., 2023). Identifying their chemical composition and main emission sources has become a priority for air quality (AQ) agencies to build up and assess efficient abatement strategies. Improving knowledge of their geographical origins is also a major challenge to better adapt local policies in a larger regional scale context.

In France, regional air quality monitoring networks (AQMN) and the national reference laboratory (termed 'LCSQA') are operating the CARA program for in situ observation of the PM chemical composition in urban environments and subsequent source apportionment studies (Favez et al., 2021). Chebaicheb et al. (2024) recently analyzed and discussed long-term (> 1 year) measurements of fine particles using online instruments at 13 CARA sites, providing $PM_1$ chemical composition, with annual mean loadings ranging from 7 to 16 µg m$^{-3}$ in French urban background environments. This concentration range is relatively low compared to other cities outside Europe but still exceeds the World Health Organization (WHO) recommended annual concentration limit of 5 µg m$^{-3}$ for $PM_{2.5}$. Organic aerosols (OA) represent a major fraction of the total $PM_1$ mass (40-60 %), a trend commonly observed worldwide (Bressi et al., 2021; Chen et al., 2022; Li et al., 2022; Via et al., 2021; Zhou et al., 2020).

Identifying the sources of the complex mix comprising the OA fraction is therefore crucial to develop effective mitigation strategies and improve AQ. Source apportionment (SA) approaches, including receptor models, have been widely used in urban AQ research during the last decades. In particular, Positive Matrix Factorization (PMF), as introduced by Paatero and Tapper (1994), stands out as one of the most extensively utilized tools (Hopke et al., 2020). PMF can be applied to various types of datasets, typically obtained from offline chemical analyses of filter samples or from online characterization of the aerosol chemistry and/or physical properties. Knowledge of OA chemistry and sources has greatly benefited from the development of aerosol mass spectrometry and subsequent application of PMF-type analysis to organic mass spectra since the mid-2000s (Crippa et al., 2014). This commonly allows to feature different families of organic compounds originating from primary emissions - typically, biomass burning OA (BBOA), hydrocarbon-like OA (HOA), cooking-like OA (COA) - or from various oxidation processes, e.g., leading to less- or more-oxidized oxygenated OA (LO-OOA and MO-OOA, respectively). Chen et al. (2022) recently proposed and applied a common protocol for advanced PMF analysis on unit-mass resolution (UMR) organic mass spectra obtained from long-term measurements at 22 European sites. This protocol is based on the use of the multi-linear engine (ME-2), allowing to introduce a priori knowledge (or assumption) on the mass spectral fingerprint of some OA factors to facilitate the comparison of SA outputs obtained at different locations. Such a standard methodology might also be of particular interest when conducting a SA study at the city scale to estimate increments due to local emissions on top of regional and/or urban background air pollution. Furthermore, previous studies also proposed to combine neighboring sites on a unique PMF analysis, in order to reinforce the consistency of the comparison of SA results obtained for each site. Such a multisite PMF analysis is considered to potentially improve the outputs' robustness, enhancing the variability in the resulting input dataset when using larger dataset than individual PMF (e.g., Pandolfi et al., 2020). To our knowledge, such a combined approach has been only applied to filter-based offline measurements, and not to aerosol mass spectrometry datasets.

In a previous paper, Chatain et al., (2021) compared the particle size distribution and aerosol concentrations between an urban background site and a roadside site during winter 2019/2020 in Strasbourg, France, showing higher particle number concentrations and particles smaller than 100 nm at the latter site compared to the former throughout the observation period. This measurement campaign also included simultaneous monitoring of black carbon (BC) and non-refractory submicron chemical species (NR-$PM_1$) at both sites, allowing for the investigation of major factors contributing to fine aerosols. In this context, this manuscript focuses on a SA study to analyze the main origins of carbonaceous species at two nearby sites located 2.5 km apart in Strasbourg. In order to compare the PMF results obtained for OA between these neighboring sites, a two-fold approach was undertaken. Initially, a standard PMF analysis was conducted independently for each site but in a harmonized way (i.e. using the same constraints and criteria). Subsequently, considering the geographical proximity of the sites, a combined PMF analysis was also carried out. Thus, the present study notably assesses the reliability and consistency of the results obtained from the individual PMF outputs compared with the combined ones. This comparative assessment also aims at discerning the main sources of pollution at these closely related sites.



## 2 Methodology

### 2.1 Sampling sites

The city of Strasbourg is located at the edge of North-Eastern France, connecting with Germany, along the river Rhine. It is part of the most populous urban area in France and the largest on a regional scale. It is highly urbanized and crossed by several major roads, including the north-south axis (A35-A4 motorway) and the east-west axis (Rhine Avenue). Residential and commercial areas are adjacent to major industrial areas to the east and south, and the entire urban area is surrounded by agricultural land.

Despite significant improvements in AQ in recent decades, Strasbourg still experiences more than ten days per year with $PM_{10}$ levels exceeding the daily limit value of 50 µg m$^{-3}$ set by the European Directive 2008/50/CE. In addition, in 2022, 100 % of the population lived in an area exceeding the WHO guideline for the annual $PM_{2.5}$ average (ATMO Grand Est, 2023). Moreover, the city can be significantly influenced by air masses from central Europe under anticyclonic conditions, as already observed for other urban areas in northern France, such as Greater Paris (MEGAPOLI, e.g., Beekmann et al. 2015; Freutel et al., 2013) and Lille (Chebaicheb et al., 2023).

A detailed description of the two sites (background and roadside) investigated here can be found in Chatain et al. (2021). Briefly, both sites correspond to fixed stations operated by the ATMO Grand Est AASQA (http://www.atmo-grandest.eu). The first one (called Danube) corresponds to an urban background station located southwest of the city center of Strasbourg (Figure 1). This station was installed at the center of a recently built eco-district between a small canal (Bassin Dusuzeau) and the Rhine Avenue. The second site (called Clemenceau) corresponds to an urban roadside station located at the corner of an intersection between two major roads in the north of the city center of Strasbourg.

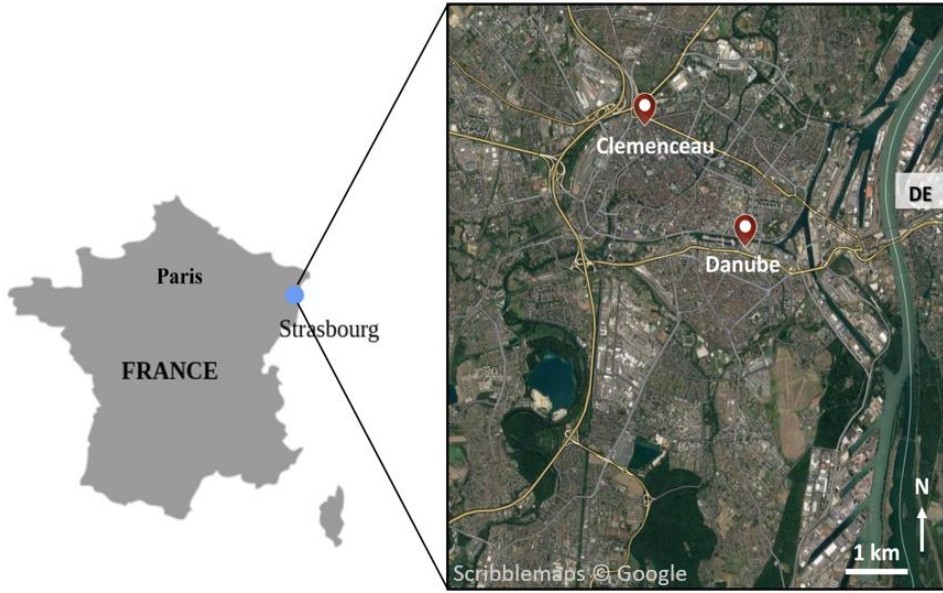

**Figure 1: Location of the two sites: Strasbourg Danube, and Strasbourg Clemenceau.**

### 2.2 Measurements

All the measurements at both sites were carried out by ATMO Grand Est. These measurements include regulatory monitoring of PM using a fine dust aerosol spectrometer (FIDAS 200, Palas Gmbh) measuring the optical light scattering of singles particles, and demonstrated to be equivalent to the gravimetric reference method for $PM_{10}$



and PM$_{2.5}$ (NF EN 12341); as well as NO$_x$ using the chemiluminescence method (APNA-370, Horiba) as
recommended by the NF EN 14211 reference method.
During winter 2019/2020, the chemical composition of NR-PM$_1$ was investigated using two quadrupole ACSMs
(Q-ACSM, *Aerosol Chemical Speciation Monitor*, Ng et al., (2011)) concomitantly at the Danube and Clemenceau
stations. In this instrument, atmospheric particles are sampled at a flow rate of 3 L min$^{-1}$ (sampling line OD = 9.5
mm; ID = 6.5 mm; 2.2 m long stainless tube) with a cut-off at 2.5 µm using a sampling head, then subsampled at
a flow rate of around 85 cc min$^{-1}$ determined by a 100 µm critical aperture mounted at the instrument inlet. The
submicron particles are then focused by an aerodynamic lens system toward a tungsten vaporizer heated at 600°C
under vacuum. The non-refractory constituents of particles are vaporized and then electronically ionized (70 eV).
The resulting fragments are separated by a quadrupole depending on their mass-to-charge ratio (m/z) operating in
a scanning mode from m/z = 10 to 150. The signal intensity (in Amps) proportional to the total amount of ions
hitting the detector (SEM: Secondary Electron Multiplier) for each m/z is then used to obtain the raw mass spectra.
The final concentrations of OA, nitrate (NO$_3$), sulfate (SO$_4$), ammonium (NH$_4$), and chloride (Cl) are obtained
using a fragmentation table (Allan et al., 2004). Roughly, inorganic compounds are first quantified based on their
fragmentation patterns, and the remaining signal at each m/z is attributed to organic fragments, forming the
measured organic fraction. The resulting OA mass spectra can then be used as an input matrix, along with its
corresponding uncertainties matrix, for PMF analysis. A critical point for the calculation of the species mass
concentrations is the determination of their ionization efficiencies. This is basically done by taking nitrate as the
reference (since it has a rather simple fragmentation pattern and few interferences at its specific *m/z* fragments)
and then measuring or assuming specific relative ionization efficiency (RIE) values relative to the NO$_3$ response
factor (RF) for other species. In the present study, RF, RIE$_{NH4}$, and RIE$_{SO4}$ were determined by on-site calibrations
during the measurement campaign, and the commonly-used default RIE values of 1.4 and 1.3 were used for OA
and Cl, respectively, for both ACSM datasets. The ACSM species were corrected using composition-dependent
collection efficiency (CDCE) correction, by applying the Middlebrook algorithm (Middlebrook et al., 2011), with
a minimum CE of 0.5.
It is also worth noting that the two ACSMs deployed here were previously compared (August-October 2019) at a
suburban background station of a nearby city (Metz-Borny, France), showing very satisfactory agreement for NO$_3$
measurements (~100%) but substantial differences - of about 30% at the highest concentration ranges - in OA
(and SO$_4$) measurements (Fig. S1). These differences did not appear to be influenced by discrepancies in relative
ion transmission (RIT) as the corresponding correction curves, based on the naphthalene internal standard
fingerprint, behaved as generally expected for ACSM devices (Fig. S2), and the highest differences were already
observed at the lowest m/z (Fig. S3). They are also unlikely to be related to differences in RIE since both
instruments were sampling the same ambient air and IE values led to a satisfactory agreement for nitrate
measurements (slope: 1.05; r² = 0.96, Fig. S1), except if the variations of organic RIE with the aerosol oxidation
state (Katz et al., 2021; Nault et al., 2023; Xu et al., 2018) might also be instrument-specific. As a matter of fact,
the few m/z ratios showing the highest concentrations for the under-estimating instrument – which was further
installed at the Clemenceau station during the wintertime Strasbourg campaign – included m/z commonly
attributed to biomass burning OA (in particular m/z 60 and 73, see Fig. S3). It should also be noted that no Pieber-
like artifact (Pieber et al., 2016) was observed during ammonium nitrate or ammonium sulfate calibrations.
Moreover, the voltage applied to the vaporizers in both instruments was kept at the values determined for each of
them by the manufacturer, theoretically ensuring a similar vaporizer temperature of about 600°C in the two
ACSM. In this context, besides any possible differences in lens transmission efficiencies, no other instrumental
bias could be suspected to explain the discrepancies observed in OA measurements during this pre-campaign
intercomparison exercise.
For the winter campaign, both Strasbourg sites were also equipped with a multi-wavelength Aethalometer (AE33,
Magee Scientific), using a sampling head with a cut-off diameter of 2.5 µm at a flow rate of 5 L/min (sampling
line OD = 12.7 mm; ID = 10.8 mm; 1.8 m long stainless tube and 2.5 m static dissipative tubing). A full description
of the AE33 operating principles is given by Drinovec et al. (2015). Briefly, it is based on the measurement of
optical attenuation in order to determine aerosol absorption coefficients (b$_{abs}$) at selected wavelengths. Aerosols
are continuously sampled onto a filter tape, causing a decrease of light transmission through the sampled filter
spot(s) which is compared to the light transmission through an unsampled area of the filter tape. In the AE33
model, optical measurements are conducted at seven optical wavelengths ranging from near-ultraviolet (UV) to
near-infrared (IR) (370, 470, 525, 590, 660, 880, and 950 nm), and sampling artifacts known as filter-loading
effect are corrected thanks to the dual-spot technology (Drinovec et al., 2015). By convention for multi-
wavelength aethalometer, equivalent Black Carbon (eBC) is derived from measurements at 880 nm, assuming a
mass absorption cross-section (MAC) value, such as:



174        $eBC = b_{abs,880nm} / MAC_{880nm}$            Eq. (1)

In line with the current ACTRIS guidelines, $b_{abs,880nm}$ was obtained applying a filter type-dependent harmonization
factor (1.76 for the MF8060 filter tape used here) to account for multiple scattering effects, and a MAC value of
7.5 $m^2$ $g^{-1}$ at 880 nm was considered to estimate eBC concentrations ([https://actris-ecac.eu/particle-light-](https://actris-ecac.eu/particle-light-absorption.html)
[absorption.html](https://actris-ecac.eu/particle-light-absorption.html)).
All these measurements underwent regular quality checks, including calibration and preventive maintenance,
following the manufacturer's recommendations, ACTRIS standard operating procedures, and guidelines provided
by ACMCC (Aerosol Chemical Monitor Calibration Center) in the ACTRIS framework (Laj et al., 2024) and by
the LCSQA at the national level. Quality control was routinely achieved following daily and weekly technical
validation procedures, supplemented by a monthly environmental validation investigation. Finally, data handling
procedures defined in Chebaicheb et al. (2024) data were applied to ACSM and AE33 datasets.
In addition, quartz fiber filters (TissuQuartz, Whatman, 47 mm diameter) pre-heated at 500°C during 4 hours and
Leckel samplers (model SEQ 47/50) running at 2.3 $m^3$/h were used to collect daily $PM_1$ samples simultaneously
at both sites from 4 to 29 February 2020 for offline analyses of organic carbon (OC) and elemental carbon (EC)
using a Sunset Lab instrument and following the EUSAAR-2 thermo-optical protocol (Cavalli et al., 2010). Daily
mean values obtained for eBC and OA (from AE33 and ACSM, respectively) were then compared to EC and OC
offline measurements, respectively. Results showed very good correlation coefficient values for both comparisons
($r^2 > 0.9$, Fig. S4) with OA-to-OC ratios (about 1.4) in the lower range of what is commonly observed in urban
environments (e.g., Aitken et al., 2008). This may be linked to the predominance of primary organic aerosols
(from various combustion processes), which are less oxidized than secondary OA (SOA). This could also be partly
related to possible OC overestimations due to positive sampling artifacts - e.g., adsorption of semi-volatile organic
compounds onto the filter (e.g., Kim et al., 2001) - and/or OA underestimation from ACSM measurements, for
instance due to poor lens transmission efficiency at the entrance of the ACSM for the finest and/or largest particles
within the submicron aerosol fraction (e.g., Liu et al., 2007). Nevertheless, the consistency obtained for OA-to-
OC ratio values with both ACSMs comforts the comparability of ACSM results.
**2.3 eBC source apportionment**
Following Sandradewi et al. (2008), multi-wavelength absorption measurements can be used to deconvolve eBC
into two main fractions, classically identified as fossil fuel ($eBC_{ff}$) and wood burning ($eBC_{wb}$) components. To do
so, it is assumed that the light absorption due to Brown Carbon (BrC) at near UV wavelengths in winter is
primarily linked to wood-burning emissions, which has been recently documented by Zhang et al. (2020) at the
national scale. More generally, the model allows distinguishing between highly efficient combustion processes
(like traffic exhaust emissions) and poor combustion conditions.
This so-called Aethalometer model is based on the additivity of absorption coefficients from both of these source
categories and on their own light absorption spectral fingerprints, such as:

208        $b_{abs,\lambda} = b_{abs,ff,\lambda} + b_{abs,wb,\lambda}$            Eq. (2)

209        $b_{abs,wb,470nm} / b_{abs,wb,950nm} = (470/950)^{-\alpha_{wb}}$            Eq. (3)

210        $b_{abs,ff,470nm} / b_{abs,ff,950nm} = (470/950)^{-\alpha_{ff}}$            Eq. (4)

where $\alpha_{ff}$ and $\alpha_{wb}$ stand for the Absorption Ångström Exponent (AAE) of the fossil fuel and wood burning
fractions, respectively. These parameters have initially been set to default values of 1 and 2, respectively
(Sandradewi et al., 2008; Drinovec et al., 2015). However, further studies illustrated that the choice of these
parameters is highly critical for the consistency of the Aethalometer model outputs so that site-specific values
should preferably be determined (e.g., Favez et al., 2010; Zotter et al., 2017; Savadkoohi et al., 2023). Following
Tobler et al. (2021), $\alpha_{ff}$ has been defined here as the first percentile of AAE values measured for ambient air
particles during the campaign, applying a stringent data point selection based on the determination coefficient ($r^2$
$> 0.99$) obtained from the fit of the $b_{abs}$ spectral dependence ($b_{abs,\lambda}$ vs. $\lambda$). Once $\alpha_{ff}$ was set for both sites (at 1.00
and 1.06 for Danube and Clemenceau, respectively), the optimal $\alpha_{wb}$ values could be investigated based on the
results of a sensitivity study aiming at optimizing correlation coefficients between $eBC_{wb}$ and $m/z$ 60 signal
(commonly used as a biomass burning tracer) from ACSM measurements while keeping the correlation between
$eBC_{ff}$ and $m/z$ 60 as low as possible. On the other hand, the correlation between $eBC_{ff}$ and NO concentration



(considered as a proxy for road traffic exhaust emission) allows to determine the minimum $\alpha_{wb}$. To be coherent,
the correlation between NO and $eBC_{ff}$ must be greater or at least equal to the correlation between NO and $eBC_{wb}$.
Such a methodology is in line with recommendations also provided by Savadkoohi et al. (2025). This led to the
determination of $\alpha_{wb}$ values of 1.6 and 1.7 for the Clemenceau and Danube sites, respectively, during the studied
period (Fig. S5).
**2.4 OA source apportionment**
The data derived from the Q-ACSM at both sites were analyzed using the Aerodyne software 'acsm local' version
6.37. Both OA concentrations matrices and their error matrices were exported to apply the PMF method using the
Source Finder Pro software (SoFi Pro v8, Datalystica Ltd., Switzerland) with the ME-2 solver within the Igor Pro
software environment (Wave Metrics, Inc., USA). Briefly, the PMF model allows for the separation of the
measured organic concentrations matrix ($x_{ij}$) at a receptor site into organic mass spectra attributed to "sources"
(profiles $f_{kj}$) and their contributions over time (time series $g_{ik}$), along with the residuals ($e_{ij}$), as described in
equation (5):
$$x_{ij} = \sum_{k=1}^{p} g_{ik} \times f_{kj} + e_{ij}$$  Eq. (5)
The objective is to find the number of factors "p" while minimizing a quantity Q defined as the sum of the squares
of the residuals ($e_{ij}$) on the measurement uncertainties ($\sigma_{ij}$):
$$Q = \sum_{i=1}^{n} \sum_{j=1}^{m} (e_{ij}/\sigma_{ij})^2$$  Eq. ( 6)
The SoFi Pro software allows the use of the a-value approach to overcome the rotational ambiguity caused by the
application of PMF. This approach helps constrain known factor profiles or time series at the site, using a scalar
a-value varying from 0 to 1, as defined in these equations:
$$f_{solution} = f_{reference} (1 \pm a)$$  Eq. (7)
$$g_{solution} = g_{reference} (1 \pm a)$$  Eq. (8)
The PMF solutions are then evaluated using the bootstrap technique which allows estimating the uncertainties of
the study.
A standard PMF analysis was first performed for each site during winter 2019/2020 (December, January,
February; DJF), as detailed in the supplementary information, section S1. Briefly, unconstrained PMF was initially
applied to pre-determine the potential number of factors (2-8 factor tests with ten PMF runs for each number of
factors), allowing to identify five main OA factors at each site, namely one oxygenated (OOA) and four primary
OA factors (hydrocarbon-like OA (HOA), cooking-like factor (COA), biomass burning OA (BBOA), as well as
a specific 58-related OA). Then, a constrained PMF was conducted using the reference profiles from Crippa et
al., (2013) for HOA and COA, which allowed us to obtain BBOA and 58-OA factors for each site without
constraining them. After establishing a reasonable PMF solution for both sites, we applied the bootstrap analysis
to test the stability of the solutions. The average bootstrapped solutions obtained at both sites are presented and
discussed in section 3.
An interesting experimental issue is the effect of possible instrumental biases - e.g., as described by Pieber et al.
(2016) - on the obtained SA results. From the comparison of PMF analyses performed on datasets simultaneously
obtained for 14 different ACSMs, Fröhlich et al. (2015) demonstrated that relatively important discrepancies in
the OA mass spectra do not necessarily lead to significant differences in the PMF results from one instrument to
another. An open question remains on the effect of mixing mass spectra datasets from two (or more) distinct
ACSMs in a single input PMF matrix. Such multisite PMF studies have been recently introduced for the combined
analysis of filter-based chemical speciation datasets, which can be obtained from offline analyses using the same
laboratory equipment (e.g., Mooibroek et al., 2011, 2016; Pandolfi et al., 2020), but have rarely, if not never, been
presented yet for online ACSM (or AMS) measurements. Considering the unexplained differences in mass spectral
fingerprints observed from co-located measurements during the preliminary intercomparison campaign in Metz
(see above), it appeared of particular interest to test here such a multisite approach combining OA measurements
at both nearby Strasbourg sites in a single PMF input matrix, also verifying the robustness and accuracy of the
individual PMF solutions. This combined PMF consisted of merging the two concentration and error matrices



from the two sites vertically with the same number of variables (m/z up to 100) and averaging the two-time series
over 30 min. As for standard independent PMF analyses, the HOA and COA factors were constrained using
Crippa's reference mass spectra. Bootstrap analysis and selection criteria were then applied to obtain the final
solution as presented in SI, section S2.

**2.5 Meteorological data and wind analysis**

Meteorological parameters have been measured at a background site located a few kilometers northwest of the
study sites. Temperature (T) and relative humidity (RH) were measured by dedicated probes (HMP Vaisala model)
and wind data (speed, WS; and direction, WD) by a wind vane (TAVID Chauvin Arnoux model). As presented
in Figure 2, the investigated period was dominated by south-western and relatively warm (5-10°C) air masses,
except during relatively short colder periods (e.g., around New Year and 22 Jan.) with northern winds.

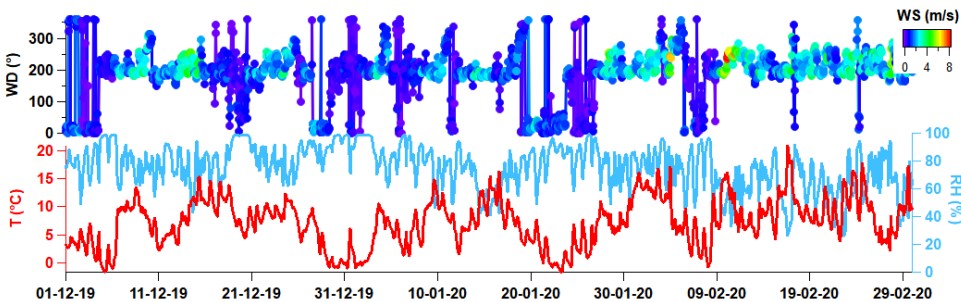

**Figure 2: Meteorological parameters in Strasbourg during winter 2019/2020.**

In order to understand the origin of air pollutants, wind and trajectory analyses were conducted, coupling pollutant
concentrations with meteorological parameters (wind speed and direction) by computing the Non-parametric
Wind Regression (NWR) model using the Zefir tool (Petit et al., 2017a), also allowing to qualitatively differentiate
between local and regional eBC and OA origins according to wind speed.

**3 Chemical composition**

Online chemical measurements could be validated by comparison with co-located regulatory $PM_{2.5}$ measurements.
In this chemical mass closure exercise, $NR-PM_1$ is first calculated as the sum of the five chemical species from
ACSM: OA, $NO_3$, $SO_4$, $NH_4$, and Cl. The $PM_1$ concentration is then approximated by adding eBC to $NR-PM_1$ and
further compared with the mass concentration of $PM_{2.5}$ measured by the FIDAS instrument at both sites. Results
indicate that ACSM and AE33 measurements together account for 62 % and 75 % of $PM_{2.5}$, with coefficients of
determination ($r^2$) equal to 0.90 and 0.87 for the Danube and Clemenceau sites, respectively (Fig. S6).
Figure 3 displays the mass concentrations of $PM_1$ species and the average contribution of eBC, $NR-PM_1$ species,
and $PM_{2.5}$ during winter 2019/2020 at the Danube and Clemenceau sites. OA dominates the average $PM_1$ chemical
composition with 48 % and 45 % at the urban background and traffic sites, respectively, as already observed in
previous studies in winter at the national scale in the Paris region (44 %), Lille (37 %) and Dunkirk (34 %)
(Chebaicheb et al., 2024, 2023, Favez et al., 2021; Zhang et al., 2021; Petit et al., 2014). Secondary inorganic
species ($NO_3$, $SO_4$, and $NH_4$) also contribute significantly, accounting for around 40 % of $PM_1$ total mass, mainly
from $NO_3$ (22-24 %). These observations are consistent with the regional formation of ammonium nitrate
($NH_4NO_3$; AN), which is greater than ammonium sulfate (($NH_4)_2SO_4$); AS). $NH_3$, considered as mainly emitted
by agricultural activities, is expected to react preferentially with sulfur compounds (mainly sulfuric acid ($H_2SO_4$)
formed from sulfur dioxide ($SO_2$)). However, regional $SO_2$ concentrations have been extremely low since the late
2010s (below 1 µg m$^{-3}$ since 2016 at regional background sites). Therefore, AS is mainly derived from long-range
transport in urban areas, and the remaining $NH_3$ is available to react with $NO_x$ present, notably to form AN locally.
However, the balance of AN formation also depends on meteorological conditions (low temperatures, high relative
humidity, high pressure), leading to higher AN concentrations in winter/early spring when these meteorological
conditions are met simultaneously with higher local $NH_3$ emissions. Previous studies conducted in this part of
Europe, e.g., in Greater Paris (Zhang et al., 2019; Petit et al., 2014) or in the Lille metropolitan area (Chebaicheb
et al., 2023) also highlighted the high contribution of organics and nitrate in $PM_1$ particles, as well as the high




impact of transboundary pollution advection from Eastern Europe in northern France for particulate matter
(Waked et al., 2018; Potier et al., 2019).
Both sites showed a similar high temporal variation, with $PM_1$ ranging from a few µg m$^{-3}$ to over 40 µg m$^{-3}$ at the
Danube site and over 60 µg m$^{-3}$ at the Clemenceau site. The coincidence of the peaks at both sites results from the
strong influence of atmospheric conditions and common local sources. The accumulation of local primary particles
is expected during the coldest periods associated with low wind speed. New Year's event is one of the peaks
associated with elevated levels due to these stable atmospheric conditions, combined with the use of fireworks
and firecrackers. In particular, some hours at the Danube site have been invalidated due to the negative chlorine
levels attributed to these particular sources, which emit chlorinated species that may be poorly and/or slowly
vaporized and not accounted for in the fragmentation table (such as chlorates, perchlorates) (Schmid et al., 2014).

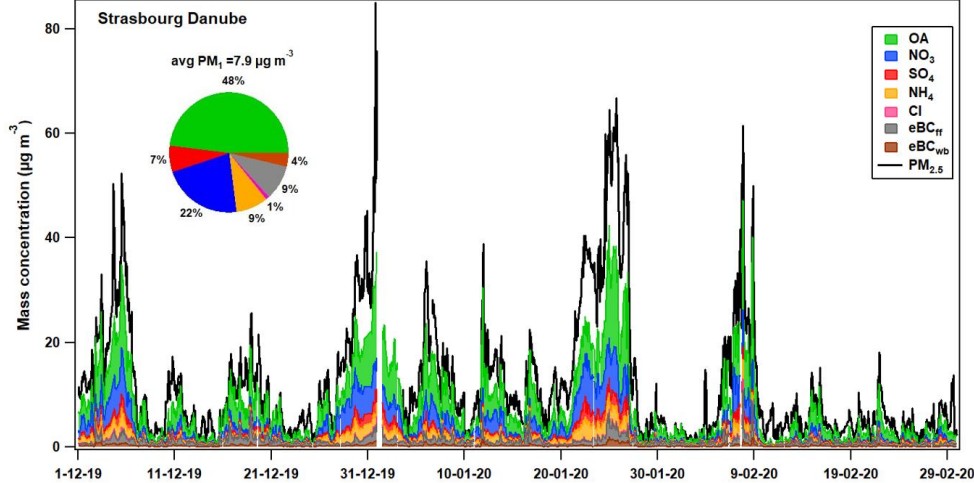


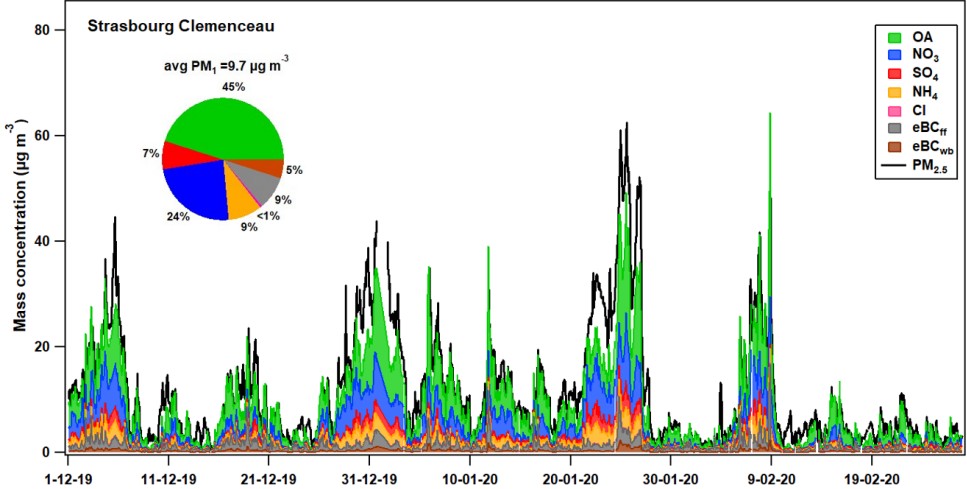


**Figure 3: $PM_1$ species at the Danube (top) and Clemenceau (bottom) sites during the studied period.**

The average mass concentrations of NR-$PM_1$ species and eBC presented in Table 1 showed only slight differences
between the two sites, with overall higher levels at the Clemenceau site. This could be attributed to the proximity
of primary exhaust and non-exhaust emissions from road traffic as well as more intense condensation and
coagulation processes. It should also be noted that the environment of the Clemenceau station is more urbanized



(city center) compared to the Danube site, which may also partly explain these observations. OA is associated
with the highest concentrations at both sites - with values of 3.8 µg m⁻³ and 4.3 µg m⁻³ at the Danube and
Clemenceau sites, respectively - reinforcing the interest in the apportionment of its main sources. The second
main compound at both sites was nitrate, with concentrations about 30% higher at Clemenceau compared to
Danube. As for OA, the difference in sulfate and $eBC_{ff}$ concentrations is about 15 % on average (with the highest
concentrations still observed at Clemenceau). Complementarily, results from offline analyses performed on filters
collected in February 2020 indicate slightly higher concentrations for Clemenceau (Table 1). Surprisingly,
however, filter-based levoglucosan analyses indicate similar concentration levels at both sites while $eBC_{wb}$
appears to be about 40 % higher at Clemenceau, and the comparison of OA mass spectra averaged over the study
period also indicates significantly higher signals for the highest m/z's, including common wood-burning tracers
(see Figure S3), at Clemenceau.
**Table 1. Average (± standard deviation) mass concentrations of PM₁ species (in µg m⁻³) at both Strasbourg sites during
the studied period.**

| Species | Danube | Clem. | Danube | Clem. | Danube | Clem. |
|---|---|---|---|---|---|---|
| | ACSM/AE33 (DJF) | | Filters (4-29 Feb.) | | ACSM/AE33 (4-29 Feb.) | |
| OA | 3.8 ± 3.6 | 4.3 ± 4.0 | | | 2.5 ± 3.1 | 3.2 ± 4.0 |
| SO₄ | 0.6 ± 0.7 | 0.7 ± 0.8 | 0.4 ± 0.3 | 0.4 ± 0.3 | 0.3 ± 0.5 | 0.45 ± 0.6 |
| NO₃ | 1.7 ± 2.1 | 2.3 ± 2.5 | 0.7 ± 1.2 | 0.8 ± 1.4 | 0.9 ± 1.5 | 1.3 ± 2.1 |
| NH₄ | 0.7 ± 0.8 | 0.9 ± 0.9 | 0.3 ± 0.5 | 0.3 ± 0.5 | 0.4 ± 0.6 | 0.5 ± 0.8 |
| eBC_ff | 0.75 ± 0.7 | 0.8 ± 0.9 | | | 0.6 ± 0.7 | 0.7 ± 1.2 |
| eBC_wb | 0.3 ± 0.3 | 0.5 ± 0.5 | | | 0.2 ± 0.3 | 0.3 ± 0.5 |
| EC | | | 0.7 ± 0.5 | 0.9 ± 0.7 | | |
| OC | | | 2.0 ± 1.7 | 2.2 ± 2.0 | | |
| Levo. | | | 0.13 ± 0.13 | 0.13 ± 0.14 | | |


**4 OA source apportionment**
Figures 4 and 5 summarize the results of the individual and combined PMF analyses for the two sites, respectively,
showing the relative contributions of each identified OA factor and their corresponding mass spectra. These results
are described in the following paragraphs, according to their nature.




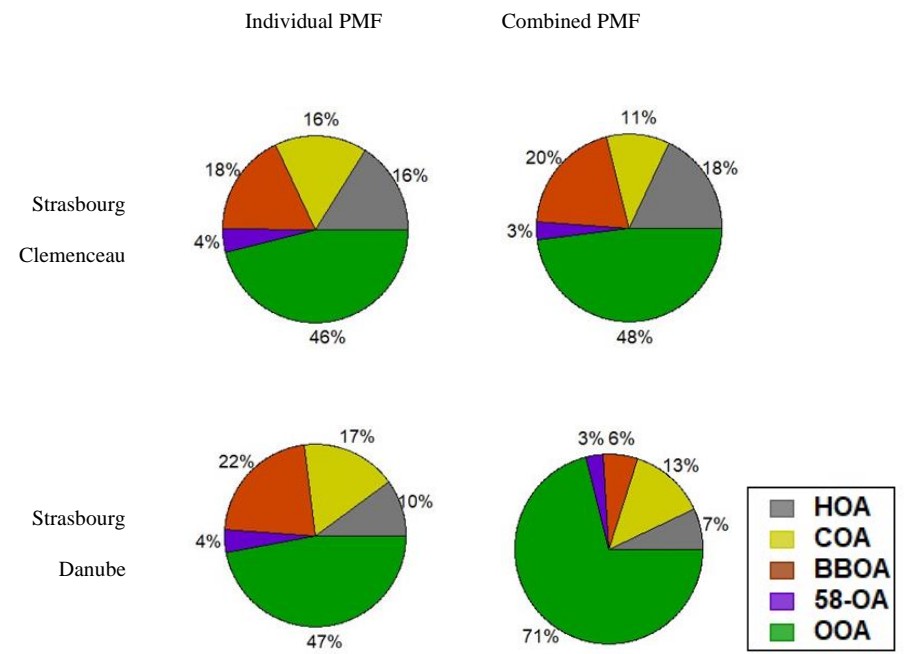

**Figure 4: Contributions of OA factors at both sites from individual and combined PMF.**

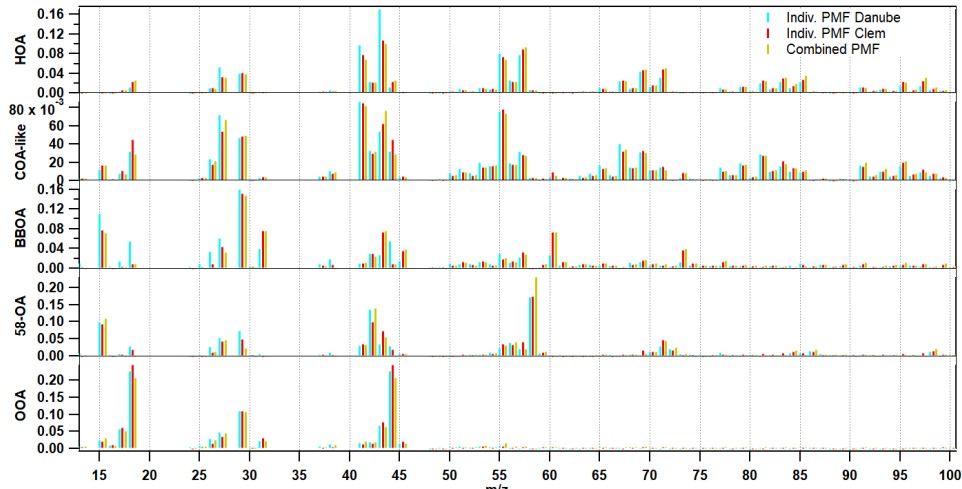


**Figure 5: Mass spectra of OA factors from individual and combined PMF for both sites.**



### 4.1 Amine-related OA

A noticeable result of the present study is the identification of a specific amine-related OA factor that was observed at both sites. This unusual factor is characterized by a high proportion of m/z 58 (mainly $C_3H_8N^+$), with a contribution of about 4 % of the total OA mass. It was observed in both unconstrained and constrained PMF analyses and regardless of the number of factors or specific runs, the unconstrained PMF consistently revealed this factor among the 4-factor (or more) solutions. It showed a relatively stable profile across different PMF runs with high contributions from m/z 58 and 42 ($C_2H_4N^+$). The time series for this factor at both sites show a highly variable temporal pattern with rather sporadic and intense peaks. The diel profile for this factor is not fully flat, displaying a slight peak in the morning and increasing towards the end of the day. However, associating this diel profile with a specific source proved to be challenging. Numerous tests were carried out in other seasons for continued measurements performed at the Danube site after the winter campaign presented in this paper. The 58-OA factor also emerged during a single study conducted in the summer, suggesting that it is not exclusively associated with winter sources.

The average mass spectrum for this factor differs from the few m/z 58-rich factors observed in previous studies based on AMS or ACSM measurements (Hildebrandt et al., 2011; Chen et al., 2021), where this factor was more likely associated with an instrument artifact. In particular, we observe high peaks for m/z 58 and 42, rather than the higher masses (m/z 84 and 98) observed in these previous studies. Furthermore, this factor does not appear when analyzing the Metz (pre-campaign) data with the same two instruments. This suggests that the factor is more closely related to local sources in Strasbourg.

The pollution rose analysis at both Strasbourg sites shows a specific and localized direction associated with this amine-related OA factor (Figure S16). The direction indicated an industrial area that could explain these specific particle emissions. A factor profile associated with amine-OA and a specific daily profile are consistent with an industrial source. This factor could therefore be associated with an industrial source of OA.

### 4.2 Other POA factors

Besides this site-specific amine-related factor, the PMF analyses identified the three main POA factors commonly observed in such a source apportionment analysis at urban sites, namely HOA, BBOA, and a COA-like factor. For HOA, the mass spectrum fingerprint is characterized by a high contribution of hydrocarbon fragments m/z 41, 43, 55, 57, 69, and 71, and its time series shows well-defined diel variations associated with morning and evening traffic peaks corresponding to home/work commutes. The BBOA factor is characterized by a high contribution at m/z 29, 60, and 73 (tracers of biomass combustion) and by a diel variation showing an increase in the evening that extends into the night, associated with residential heating in winter. Some differences were observed between the BBOA profiles at the two sites. These differences can be explained by particle aging, especially with a higher 44/60 ratio at the Danube site than at the Clemenceau site, indicating more oxidized particles from biomass burning at the Danube site. Compared to HOA and BBOA, COA profiles are usually characterized by higher peaks at m/z 41 and 55, and by diel cycles showing midday (lunch) and evening (dinner) peaks associated with cooking activities. Such a clear diel profile could not be obtained for the Danube site, where it shows variations quite similar to those of BBOA, with a concentration maximum in the late evening. Therefore, we rather refer to this factor as COA-like.

For individual PMF outputs, HOA, BBOA, and COA-like contributed 16 %, 16 %, and 18 % of the organic matter, respectively, at the Clemenceau site. A lower HOA contribution (10 %) was observed for Danube, consistent with an increased influence of traffic at the Clemenceau site. The two primary factors HOA and BBOA showed a strong correlation with $eBC_{ff}$ and $eBC_{wb}$ ($r^2$ (HOA, $eBC_{ff}$) = 0.81 and 0.73; and $r^2$ (BBOA, $eBC_{wb}$) = 0.92 and 0.90 for the Clemenceau and Danube sites, respectively, Figure S13, SI). COA-like at the Danube site has a fairly similar contribution to COA-like at the Clemenceau site (about 17 %) but with a lower mass concentration of 0.6 µg m⁻³ compared to Clemenceau (0.9 µg m⁻³) which is closer to the city center. However, BBOA shows a higher contribution at the Danube site (22 % of the total OA mass) with a concentration of 0.8 µg m⁻³, close to that of Clemenceau which could be explained by the differences in the factor profile. A comparison of m/z 60 at Danube and Clemenceau (Figure S14 in the SI) shows an m/z 60 signal twice as high at Clemenceau as at Danube.

The results of the combined PMF analysis may help to gain a deeper understanding of the variation of these POA factors between the two sites. Interestingly, the contributions obtained from the combined PMF differ from those obtained using the individual PMF for Danube, whereas only minor differences are obtained for Clemenceau, with slightly more HOA and BBOA and less COA-like at this site (Figure 5). For the Danube site, however, the relative



contributions and mass concentrations of these factors are significantly altered, especially for HOA and BBOA.
The COA-like factor shows similar mass concentrations close to 0.6 µg m$^{-3}$, representing 17 % and 13 % of the
total mass for the individual and combined PMF, respectively. HOA also decreased slightly from 10 % to 7 %
between the two PMF approaches. A strong decrease in BBOA is observed, from 22 % to 6 %, consistent with
the m/z 60 comparison showing less fresh BBOA at the Danube site. In fact, the BBOA profile for the combined
PMF is more similar to the BBOA profile for the individual PMF at the Clemenceau site compared to the Danube
site.
**4.3 Oxygenated organic aerosols (OOA)**
The obtained OOA mass spectra are consistent with those reported in the literature, associated with a stable diel
variation, and characterized by a high contribution of m/z 44 (Figure S10). At both sites, the OOA factor showed
similar profiles and diel variations. For the individual PMF, OOA dominated OA at both sites with a contribution
of about 47 % but with different mass concentrations of 2.6 and 1.7 µg m$^{-3}$ at the Clemenceau and Danube sites,
respectively.
The combined PMF identified two OOA factors, named OOA1 and OOA2 (section S2, Figure S9). Because the
two factors are correlated, we summed them together to obtain a single OOA factor (Figure S11). At the
Clemenceau site, the contribution of the OOA factor (48 %) is similar to the one obtained from the individual
PMF, with a mass concentration of 2.8 µg m$^{-3}$. At the Danube site, the contribution of the OOA factor strongly
increased from 47 % for the individual PMF to 71 % for the combined one (Figure 4), reaching a mass
concentration of 2.6 µg m$^{-3}$, close to that of the Clemenceau site. These combined PMF results provided further
insight into this factor and demonstrated its regional origin. In Clemenceau, we observed a balance between
primary and secondary OA factors, while for Danube, the combined PMF results indicated an average OOA
contribution of about 70 % of the total OA.

**5 Discussion and concluding remarks**
The present study provides an opportunity to qualitatively assess the measurement consistency and results of
carbonaceous aerosol source apportionment analyses conducted with the same type of instrument at two
neighboring urban sites. The two ACSMs used in this study to characterize non-refractory submicron chemical
species were first compared at the same location prior to the campaign. This side-by-side comparison showed a
very good agreement for nitrate concentrations, confirming the consistency of the response factors obtained from
the calibration of both ACSMs. However, significant discrepancies - of about 30 % - were obtained for OA (and
sulfate) mass concentrations, and the comparison of OA mass spectra showed a surprising behavior with a few
*m/z's* (including *m/z* 60 and 73, commonly used as biomass burning tracers) at higher levels in the instrument
which presented overall lower loadings in total OA concentrations. Quality assurance and quality checks were
performed according to the most advanced recommendations provided by the manufacturer and the scientific
community, and no specific instrumental bias could be identified to explain these differences. Therefore, both
ACSMs can be considered as running under their usual and proper operating conditions, with discrepancies mainly
due to inherent technical specificities of each instrument. It should be noted that the two ACSMs do not have the
same age or date of manufacture (the Clemenceau ACSM dates from 2014 and the Danube ACSM from 2018),
which may influence the evolution of the instrument over time.
Once installed at their respective monitoring stations in Strasbourg, the ACSMs provided meaningful
measurements of the submicron aerosol chemical species, with moderately higher concentrations at the
Clemenceau traffic site compared to the urban background Danube station, which can be attributed to more intense
primary emissions and/or transformation processes at the roadside. Filter-based offline measurements available
for the last month of the campaign tend to confirm these observations, with slightly higher concentrations of major
chemical species at Clemenceau. Results from individual PMF analyses also indicated higher HOA contributions
for Clemenceau, in good agreement with moderately higher eBC$_{ff}$ and EC concentrations at the same site
compared to Danube. They also pointed to similar BBOA contributions at the two stations, in agreement with the
comparison of filter-based levoglucosan measurements. Less expected is the significantly higher eBC$_{wb}$ loading
obtained from the application of the so-called Aethalometer model to AE33 measurements at both sites. This may
be due to the choice of site-specific sets of AAE values ($\alpha_{ff}$ and $\alpha_{wb}$) chosen to represent the two eBC subfractions
at each monitoring site. This result confirms the high sensitivity of the Aethalometer model to the empirical



assumptions to be made for its application. In particular, this may illustrate how cautiously it should be considered
for any eBC source apportionment study at a traffic site (e.g., Savadkoohi et al., 2023).
The COA-like factor showed the expected peaks at lunch and dinner time for Clemenceau, but also a diel cycle
relatively similar to BBOA (with no significant midday increase) for Danube, illustrating the difficulty in
attributing a pure cooking origin to this factor. Interestingly, the results of the combined PMF analysis may
improve the apportionment of these cooking emissions, reducing the COA-to-HOA ratio at night - especially for
Clemenceau - compared to individual PMF outputs (Fig. S12). However, such a combined PMF analysis may not
be suitable to improve the consistency of other OA factors, probably due to instrumental specificities leading to
differences in the OA mass spectra obtained by the two instruments. More specifically, the OA profiles obtained
by the combined PMF method exhibit a greater similarity to the profiles obtained by the individual PMF analysis
at Clemenceau than to those obtained by the individual PMF analysis at Danube. As a result, there is a reduced
contribution of m/z 43 (associated with HOA) and m/z 44 (associated with BBOA) at Danube when applying the
combined PMF approach as opposed to the individual PMF analyses. This discrepancy leads to an underestimation
of these factors (HOA and BBOA) and an overestimation of OOA at the Danube site when using the combined
PMF method.
Finally, simultaneous ACSM measurements at these paired sites allow to confirm the existence and substantial
influence of an amine-related OA factor in Strasbourg. This unique factor, characterized by a high proportion of
m/z 58, represents approximately 4 % of the total OA mass and is consistently observed in both PMF analyses.
The diel profile of this factor shows peaks in the morning and late in the day, but its specific source remains
challenging to identify. It differs from previously observed m/z 58-rich factors, suggesting a distinct local source,
possibly related to industrial emissions. Further investigation is needed to determine the exact source of this
intriguing amine-related OA factor.
In summary, the comparison of different PMF methods carried out in this study highlights caveats and limitations
inherent to such kind of SA approach First, the elucidation of OA sources based on factors derived from PMF
should be interpreted with caution considering real-world . In addition, attention should be exercised when
combining data from different measurement instruments, as they are not strictly identical in terms of sensitivity.
However, positioning two instruments in the same location (or close to each other) can help to verify the presence
of atypical or unique factors and explain discrepancies. These limitations introduce uncertainties in the
apportionment of OA sources and in the consistency of factor interpretation. In order to improve the identification
and interpretation of PMF factors, we propose the integration of complementary datasets (e.g., molecular tracers),
which would provide additional constraints. Future work should include a focus on refined methodologies to better
handle multi-instrument and multi-timescale datasets, and on the elaboration of standardized protocols for inter-
instrument comparisons. Ultimately, improving these methods will lead to a better understanding of the sources,
evolution, and role of OA in the atmosphere, which is crucial for accurately assessing their impacts on air quality,
health, and climate.

**Data availability**
Data for Strasbourg Danube are available at https://doi.org/10.5281/zenodo.13318298 (Chebaicheb et al., 2024).
Data for Strasbourg Clemenceau are available at https://doi.org/10.5281/zenodo.14855186 (Chebaicheb et al.,
2025). More details on the analyses are available upon request to the contact author Hasna Chebaicheb
(hasna.chebaicheb@ineris.fr).
**Author contributions**
HC: Data curation, Formal analysis, Investigation, Methodology, Visualisation, Conceptualization, Writing -
Original Draft
MC: Data curation, Formal analysis, Visualisation, Conceptualization, Resources, Writing - Original Draft
OF: Conceptualization, Methodology, Validation, Supervision, Writing - Original Draft, Project administration,
Funding acquisition



JB: Conceptualization, Validation, Supervision, Writing - Review & Editing
VC: Conceptualization, Writing - Review & Editing
TA: Conceptualization, Writing - Review & Editing
MG: Formal analysis, Writing - Review & Editing
EJ: Conceptualization, Writing - Review & Editing
CM: Conceptualization, Supervision, Writing - Review & Editing
VR: Conceptualization, Validation, Supervision, Writing - Review & Editing, Project administration, Funding
acquisition
**Acknowledgments**
IMT Nord Europe and INERIS participated in the COST COLOSSAL Action CA16109.
**Funding**
H. Chebaicheb's PhD grant was supported by the LCSQA funded by the French Ministry of Environment. IMT
Nord Europe also acknowledges financial support from the Labex CaPPA project, which is funded by the French
National Research Agency (ANR) through the PIA (Programme d'Investissement d'Avenir) under contract ANR-
11-LABX-0005-01.
**Conflicts of Interest.** The authors declare no conflict of interest.

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

Background Measurements of Submicron Aerosol Number Concentration and Size Distribution (in the Range 20–
800 nm), along with Chemical Composition in Strasbourg, France, Atmosphere, 12, 71,
https://doi.org/10.3390/atmos12010071, 2021.
Chebaicheb, H., F. de Brito, J., Chen, G., Tison, E., Marchand, C., Prévôt, A. S. H., Favez, O., and Riffault, V.:
Investigation of four-year chemical composition and organic aerosol sources of submicron particles at the ATOLL
site in northern France, Environ. Pollut., 330, 121805, https://doi.org/10.1016/j.envpol.2023.121805, 2023.
Chebaicheb, H., de Brito, J. F., Amodeo, T., Couvidat, F., Petit, J.-E., Tison, E., Abbou, G., Baudic, A., Chatain,
M., Chazeau, B., Marchand, N., Falhun, R., Francony, F., Ratier, C., Grenier, D., Vidaud, R., Zhang, S., Gille, G.,
Meunier, L., Marchand, C., Riffault, V., and Favez, O.: Multiyear high-temporal-resolution measurements of
submicron aerosols at 13 French urban sites: data processing and chemical composition, Earth Syst. Sci. Data, 16,
5089–5109, https://doi.org/10.5194/essd-16-5089-2024, 2024.
Chebaicheb, H., Ferreira de Brito, J., Amodeo, T., Couvidat, F., Petit, J.-E., Tison, E., Abbou, G., Alexia, B.,
Chatain, M., Chazeau, B., Marchand, N., Falhun, R., Francony, F., Ratier, C., Grenier, D., Vidaud, R., Zhang, S.,
Gille, G., Meunier, L., Marchand, C., Riffault, V., and Favez, O.: Multi-year high time resolution measurements
of fine PM at 13 sites of the French Operational Network (CARA program), In Earth System Science Data, Zenodo
[data set], https://doi.org/10.5281/zenodo.13318298, 2024.
Chebaicheb, H., Chatain, M., Favez, O., Ferreira de Brito, J., Crenn, V., Amodeo, T., Gherras, M., Jantzem, E.,
Marchand, C., & Riffault, V. (2025). Lessons learned from the comparison and combination of fine carbonaceous
aerosol source apportionment at two locations in the city of Strasbourg, France [Data set]. In Atmospheric
Chemistry and Physics (ACP). Zenodo. https://doi.org/10.5281/zenodo.14855186.
Chen, G., Sosedova, Y., Canonaco, F., Fröhlich, R., Tobler, A., Vlachou, A., Daellenbach, K. R., Bozzetti, C.,
Hueglin, C., Graf, P., Baltensperger, U., Slowik, J. G., El Haddad, I., and Prévôt, A. S. H.: Time-dependent source
apportionment of submicron organic aerosol for a rural site in an alpine valley using a rolling positive matrix
factorization (PMF) window, Atmospheric Chem. Phys., 21, 15081–15101, https://doi.org/10.5194/acp-21-
567   15081-2021, 2021.

Chen, G., Canonaco, F., Tobler, A., Aas, W., Alastuey, A., Allan, J., Atabakhsh, S., Aurela, M., Baltensperger,
U., Bougiatioti, A., De Brito, J. F., Ceburnis, D., Chazeau, B., Chebaicheb, H., Daellenbach, K. R., Ehn, M., El
Haddad, I., Eleftheriadis, K., Favez, O., Flentje, H., Font, A., Fossum, K., Freney, E., Gini, M., Green, D. C.,
Heikkinen, L., Herrmann, H., Kalogridis, A.-C., Keernik, H., Lhotka, R., Lin, C., Lunder, C., Maasikmets, M.,
Manousakas, M. I., Marchand, N., Marin, C., Marmureanu, L., Mihalopoulos, N., Močnik, G., Nęcki, J., O'Dowd,
C., Ovadnevaite, J., Peter, T., Petit, J.-E., Pikridas, M., Matthew Platt, S., Pokorná, P., Poulain, L., Priestman, M.,
Riffault, V., Rinaldi, M., Różański, K., Schwarz, J., Sciare, J., Simon, L., Skiba, A., Slowik, J. G., Sosedova, Y.,
Stavroulas, I., Styszko, K., Teinemaa, E., Timonen, H., Tremper, A., Vasilescu, J., Via, M., Vodička, P.,
Wiedensohler, A., Zografou, O., Cruz Minguillón, M., and Prévôt, A. S. H.: European aerosol phenomenology −
8: Harmonised source apportionment of organic aerosol using 22 Year-long ACSM/AMS datasets, Environ. Int.,
166, 107325, https://doi.org/10.1016/j.envint.2022.107325, 2022.
Crippa, M., DeCarlo, P. F., Slowik, J. G., Mohr, C., Heringa, M. F., Chirico, R., Poulain, L., Freutel, F., Sciare,
J., Cozic, J., Di Marco, C. F., Elsasser, M., Nicolas, J. B., Marchand, N., Abidi, E., Wiedensohler, A., Drewnick,
F., Schneider, J., Borrmann, S., Nemitz, E., Zimmermann, R., Jaffrezo, J.-L., Prévôt, A. S. H., and Baltensperger,
U.: Wintertime aerosol chemical composition and source apportionment of the organic fraction in the metropolitan
area of Paris, Atmospheric Chem. Phys., 13, 961–981, https://doi.org/10.5194/acp-13-961-2013, 2013.
Drinovec, L., Močnik, G., Zotter, P., Prévôt, A. S. H., Ruckstuhl, C., Coz, E., Rupakheti, M., Sciare, J., Müller,
T., Wiedensohler, A., and Hansen, A. D. A.: The "dual-spot" Aethalometer: an improved measurement of aerosol
black carbon with real-time loading compensation, Atmospheric Meas. Tech., 8, 1965–1979,
https://doi.org/10.5194/amt-8-1965-2015, 2015.



588 Favez, O., Weber, S., Petit, J.-E., Alleman, L., Albinet, A., Riffault, V., Chazeau, B., Amodeo, T., Salameh, D.,
589 Zhang, Y., Srivastava, D., Samaké, A., Aujay, R., Papin, A., Bonnaire, N., Boullanger, C., Chatain, M., Chevrier,
590 F., Detournay, A., and Leoz-Garziandia, E.: Overview of the French Operational Network for In Situ Observation
591 of PM Chemical Composition and Sources in Urban Unvironments (CARA Program),
592 https://doi.org/10.20944/preprints202101.0182.v1, 2021.

593 Guide méthodologique pour la mesure des concentrations en ammoniac dans l'air ambiant | LCSQA:
594 https://www.lcsqa.org/fr/rapport/guide-methodologique-pour-la-mesure-des-concentrations-en-ammoniac-dans-
595 lair-ambiant, last access: 31 July 2023.

596 Hildebrandt, L., Kostenidou, E., Lanz, V. A., Prevot, A. S. H., Baltensperger, U., Mihalopoulos, N., Laaksonen,
597 A., Donahue, N. M., and Pandis, S. N.: Sources and atmospheric processing of organic aerosol in the
598 Mediterranean: insights from aerosol mass spectrometer factor analysis, Atmospheric Chem. Phys., 11, 12499–
599 12515, https://doi.org/10.5194/acp-11-12499-2011, 2011.

600 Hopke, P. K., Dai, Q., Li, L., and Feng, Y.: Global review of recent source apportionments for airborne particulate
601 matter, Sci. Total Environ., 740, 140091, https://doi.org/10.1016/j.scitotenv.2020.140091, 2020.

602 Katz, E. F., Guo, H., Campuzano-Jost, P., Day, D. A., Brown, W. L., Boedicker, E., Pothier, M., Lunderberg, D.
603 M., Patel, S., Patel, K., Hayes, P. L., Avery, A., Hildebrandt Ruiz, L., Goldstein, A. H., Vance, M. E., Farmer, D.
604 K., Jimenez, J. L., and DeCarlo, P. F.: Quantification of cooking organic aerosol in the indoor environment using
605 aerodyne aerosol mass spectrometers, Aerosol Science and Technology, 55, 1099–1114,
606 https://doi.org/10.1080/02786826.2021.1931013, 2021.

607 Kim, B.M., Cassmassi, J., Hogo, H., Zeldin, M.D., 2001. Positive Organic Carbon Artifacts on Filter Medium
608 During PM2.5 Sampling in the South Coast Air Basin. Aerosol Science and Technology 34, 35–41.
609 https://doi.org/10.1080/02786820118227.

610 Laj, P., Myhre, C.L., Riffault, V., Amiridis, V., Fuchs, H., Eleftheriadis, K., Petäjä, T., Salameh, T., Kivekäs, N.,
611 Juurola, E., Saponaro, G., Philippin, S., Cornacchia, C., Arboledas, L.A., Baars, H., Claude, A., Mazière, M.D.,
612 Dils, B., Dufresne, M., Evangeliou, N., Favez, O., Fiebig, M., Haeffelin, M., Herrmann, H., Höhler, K., Illmann,
613 N., Kreuter, A., Ludewig, E., Marinou, E., Möhler, O., Mona, L., Murberg, L.E., Nicolae, D., Novelli, A.,
614 O'Connor, E., Ohneiser, K., Altieri, R.M.P., Picquet-Varrault, B., Pinxteren, D. van, Pospichal, B., Putaud, J.-P.,
615 Reimann, S., Siomos, N., Stachlewska, I., Tillmann, R., Voudouri, K.A., Wandinger, U., Wiedensohler, A.,
616 Apituley, A., Comerón, A., Gysel-Beer, M., Mihalopoulos, N., Nikolova, N., Pietruczuk, A., Sauvage, S., Sciare,
617 J., Skov, H., Svendby, T., Swietlicki, E., Tonev, D., Vaughan, G., Zdimal, V., Baltensperger, U., Doussin, J.-F.,
618 Kulmala, M., Pappalardo, G., Sundet, S.S., Vana, M., 2024. Aerosol, Clouds and Trace Gases Research
619 Infrastructure (ACTRIS): The European Research Infrastructure Supporting Atmospheric Science. Bulletin of the
620 American Meteorological Society 105, E1098–E1136. https://doi.org/10.1175/BAMS-D-23-0064.1.

621 Li, S., Chen, C., Yang, G., Fang, J., Sun, Y., Tang, L., Wang, H., Xiang, W., Zhang, H., Croteau, P.L., Jayne, J.T.,
622 Liao, H., Ge, X., Favez, O., Zhang, Y., 2022. Sources and processes of organic aerosol in non-refractory $PM_1$ and
623 $PM_{2.5}$ during foggy and haze episodes in an urban environment of the Yangtze River Delta, China. Environmental
624 Research 212, 113557. https://doi.org/10.1016/j.envres.2022.113557.

625 Liu, P.S.K., Deng, R., Smith, K.A., Williams, L.R., Jayne, J.T., Canagaratna, M.R., Moore, K., Onasch, T.B.,
626 Worsnop, D.R., Deshler, T., 2007. Transmission Efficiency of an Aerodynamic Focusing Lens System:
627 Comparison of Model Calculations and Laboratory Measurements for the Aerodyne Aerosol Mass Spectrometer.
628 Aerosol Science and Technology 41, 721–733. https://doi.org/10.1080/02786820701422278.

629 Masson-Delmotte, V., Zhai, P., Pirani, A., Connors, S. L., Péan, C., Berger, S., Caud, N., Chen, Y., Goldfarb, L.,
630 Gomis, M. I., Huang, M., Leitzell, K., Lonnoy, E., Matthews, J. B. R., Maycock, T. K., Waterfield, T., Yelekçi,
631 Ö., Yu, R., and Zhou, B. (Eds.): Climate Change 2021: The Physical Science Basis. Contribution of Working
632 Group I to the Sixth Assessment Report of the Intergovernmental Panel on Climate Change, Cambridge University
633 Press, Cambridge, United Kingdom and New York, NY, USA, https://doi.org/10.1017/9781009157896, 2021.



Middlebrook, A.M., Bahreini, R., Jimenez, J.L., Canagaratna, M.R., 2011. Evaluation of Composition-Dependent
Collection Efficiencies for the Aerodyne Aerosol Mass Spectrometer using Field Data. Aerosol Science and
Technology 46, 258–271. https://doi.org/10.1080/02786826.2011.620041.
Mooibroek, D., Schaap, M., Weijers, E. P., and Hoogerbrugge, R.: Source apportionment and spatial variability
of PM$_{2.5}$ using measurements at five sites in the Netherlands, Atmos. Environ., 45, 4180–4191,
https://doi.org/10.1016/j.atmosenv.2011.05.017, 2011.
Mooibroek, D., Staelens, J., Cordell, R., Panteliadis, P., Delaunay, T., Weijers, E., Vercauteren, J., Hoogerbrugge,
R., Dijkema, M., Monks, P. S., and Roekens, E.: PM$_{10}$ Source Apportionment in Five North Western European
Cities—Outcome of the Joaquin Project, https://doi.org/10.1039/9781782626589-00264, 2016.
Nault, B. A., Croteau, P., Jayne, J., Williams, A., Williams, L., Worsnop, D., Katz, E. F., DeCarlo, P. F., and
Canagaratna, M.: Laboratory evaluation of organic aerosol relative ionization efficiencies in the aerodyne aerosol
mass spectrometer and aerosol chemical speciation monitor, Aerosol Science and Technology, 57, 981–997,
https://doi.org/10.1080/02786826.2023.2223249, 2023.
Ng, N. L., Herndon, S. C., Trimborn, A., Canagaratna, M. R., Croteau, P. L., Onasch, T. B., Sueper, D., Worsnop,
D. R., Zhang, Q., Sun, Y. L., and Jayne, J. T.: An Aerosol Chemical Speciation Monitor (ACSM) for Routine
Monitoring of the Composition and Mass Concentrations of Ambient Aerosol, Aerosol Sci. Technol., 45, 780–
794, https://doi.org/10.1080/02786826.2011.560211, 2011.
Paatero, P., Tapper, U., 1994. Positive matrix factorization: a non-negative factor model with optimal utilization
of error estimates of data values. Environmetrics 5, 111–126. https://doi.org/10.1002/env.3170050203.
Pandolfi, M., Mooibroek, D., Hopke, P., van Pinxteren, D., Querol, X., Herrmann, H., Alastuey, A., Favez, O.,
Hüglin, C., Perdrix, E., Riffault, V., Sauvage, S., van der Swaluw, E., Tarasova, O., Colette, A., 2020. Long-range
and local air pollution: what can we learn from chemical speciation of particulate matter at paired sites?
Atmospheric Chemistry and Physics 20, 409–429. https://doi.org/10.5194/acp-20-409-2020.
Petit, J.-E., Favez, O., Sciare, J., Canonaco, F., Croteau, P., Močnik, G., Jayne, J., Worsnop, D., and Leoz-
Garziandia, E.: Submicron aerosol source apportionment of wintertime pollution in Paris, France by double
positive matrix factorization (PMF$^2$) using an aerosol chemical speciation monitor (ACSM) and a multi-
wavelength Aethalometer, Atmospheric Chem. Phys., 14, 13773–13787, https://doi.org/10.5194/acp-14-13773-
661   2014, 2014.

Petit, J.-E., Pallares, C., Favez, O., Alleman, L., Bonnaire, N., and Rivière, E.: Sources and Geographical Origins
of PM$_{10}$ in Metz (France) Using Oxalate as a Marker of Secondary Organic Aerosols by Positive Matrix
Factorization Analysis, Atmosphere, 10, 370, https://doi.org/10.3390/atmos10070370, 2019.
Potier, E., Waked, A., Bourin, A., Minvielle, F., Péré, J.C., Perdrix, E., Michoud, V., Riffault, V., Alleman, L.Y.,
Sauvage, S., 2019. Characterizing the regional contribution to PM10 pollution over northern France using two
complementary approaches: Chemistry transport and trajectory-based receptor models. Atmospheric Research
223, 1–14. https://doi.org/10.1016/j.atmosres.2019.03.002
Qi, L., Bozzetti, C., Corbin, J. C., Daellenbach, K. R., El Haddad, I., Zhang, Q., Wang, J., Baltensperger, U.,
Prévôt, A. S. H., Chen, M., Ge, X., and Slowik, J. G.: Source identification and characterization of organic nitrogen
in atmospheric aerosols at a suburban site in China, Sci. Total Environ., 818, 151800,
https://doi.org/10.1016/j.scitotenv.2021.151800, 2022.
Sandradewi, J., Prevot, A., Weingartner, E., Schmidhauser, R., Gysel, M., and Baltensperger, U.: A study of wood
burning and traffic aerosols in an Alpine valley using a multi-wavelength Aethalometer, Atmos. Environ., 42,
101–112, https://doi.org/10.1016/j.atmosenv.2007.09.034, 2008.
Savadkoohi, M., et al., Addressing the advantages and limitations of using Aethalometer data to determine the
optimal Absorption Ångström Exponents (AAEs) values for eBC source apportionment, 2025, submitted to
Environment International.



Schmid, P., Bogdal, C., Wang, Z., Azara, V., Haag, R., and von Arx, U.: Releases of chlorobenzenes,
chlorophenols    and    dioxins    during    fireworks,    Chemosphere,    114,    158–164,
https://doi.org/10.1016/j.chemosphere.2014.03.088, 2014.
Tobler, A.K., Skiba, A., Canonaco, F., Močnik, G., Rai, P., Chen, G., Bartyzel, J., Zimnoch, M., Styszko, K.,
Nęcki, J., Furger, M., Różański, K., Baltensperger, U., Slowik, J.G., Prevot, A.S.H., 2021. Characterization of
non-refractory (NR) PM$_1$ and source apportionment of organic aerosol in Kraków, Poland. Atmospheric
Chemistry and Physics 21, 14893–14906. https://doi.org/10.5194/acp-21-14893-2021.
Via, M., Minguillón, M.C., Reche, C., Querol, X., Alastuey, A., 2021. Increase in secondary organic aerosol in
an urban environment. Atmospheric Chemistry and Physics 21, 8323–8339. https://doi.org/10.5194/acp-21-8323-
688    2021.

Waked, A., Bourin, A., Michoud, V., Perdrix, E., Alleman, L.Y., Sauvage, S., Delaunay, T., Vermeesch, S., Petit,
J.-E., Riffault, V., 2018. Investigation of the geographical origins of PM10 based on long, medium and short-
range air mass back-trajectories impacting Northern France during the period 2009–2013. Atmospheric
Environment 193, 143–152. https://doi.org/10.1016/j.atmosenv.2018.08.015
WHO    Air    Quality    Guidelines:    https://www.c40knowledgehub.org/s/article/WHO-Air-Quality-
Guidelines?language=en_US, last access: 23 January 2023.
Xu, W., Lambe, A., Silva, P., Hu, W., Onasch, T., Williams, L., Croteau, P., Zhang, X., Renbaum-Wolff, L.,
Fortner, E., Jimenez, J. L., Jayne, J., Worsnop, D., and Canagaratna, M.: Laboratory evaluation of species-
dependent relative ionization efficiencies in the Aerodyne Aerosol Mass Spectrometer, Aerosol Sci. Technol., 52,
626–641, https://doi.org/10.1080/02786826.2018.1439570, 2018.
Zhang, S., Tison, E., Dusanter, S., Beaugard, C., Gengembre, C., Augustin, P., Fourmentin, M., Delbarre, H.,
Riffault, V., 2021. Near real-time PM1 chemical composition measurements at a French urban background and
coastal site under industrial influence over more than a year: Temporal variability and assessment of sulfur-
containing emissions. Atmospheric Environment 244, 117960. https://doi.org/10.1016/j.atmosenv.2020.117960
Zhang, Y., Favez, O., Petit, J.-E., Canonaco, F., Truong, F., Bonnaire, N., Crenn, V., Amodeo, T., Prévôt, A. S.
H., Sciare, J., Gros, V., and Albinet, A.: Six-year source apportionment of submicron organic aerosols from near-
continuous highly time-resolved measurements at SIRTA (Paris area, France), Atmospheric Chem. Phys., 19,
14755–14776, https://doi.org/10.5194/acp-19-14755-2019, 2019.
Zhang, Y., Albinet, A., Petit, J.-E., Jacob, V., Chevrier, F., Gille, G., Pontet, S., Chrétien, E., Dominik-Sègue, M.,
Levigoureux, G., Močnik, G., Gros, V., Jaffrezo, J.-L., Favez, O., 2020. Substantial brown carbon emissions from
wintertime    residential    wood    burning    over    France.    Science    of    The    Total    Environment    743,    140752.
https://doi.org/10.1016/j.scitotenv.2020.140752.
Zhou, S., Collier, S., Xu, J., Mei, F., Wang, J., Lee, Y.-N., Sedlacek III, A. J., Springston, S. R., Sun, Y., and
Zhang, Q.: Influences of upwind emission sources and atmospheric processing on aerosol chemistry and properties
at    a    rural    location    in    the    Northeastern    U.S.,    J.    Geophys.    Res.    Atmospheres,    121,    6049–6065,
https://doi.org/10.1002/2015JD024568, 2016.
Zhou, W., Xu, W., Kim, H., Zhang, Q., Fu, P., Worsnop, D.R., Sun, Y., 2020. A review of aerosol chemistry in
Asia: insights from aerosol mass spectrometer measurements. Environ. Sci.: Processes Impacts 22, 1616–1653.
https://doi.org/10.1039/D0EM00212G.