# Peer review of "Measurement report: Lessons learned from the comparison"

_EGUsphere, 2025_

## Author Comment (AC1)

**Response to the reviewers on EGUSPHERE-2025-648 "Measurement report: Lessons learned from the comparison and combination of fine carbonaceous aerosol source apportionment at two locations in the city of Strasbourg, France"**

We thank the editor and the reviewers for their constructive advice and comments on our manuscript. In the following, we respond to all reviewers' comments using a black font for original review comments, green font for authors' responses, and blue font for changes in the revised version.

**#Referee 2**

Chebaicheb et al. present a comparison of aerosol composition/concentration and positive matrix factorization (PMF) results from two quadrupole aerosol chemical speciation monitors (Q-ACSMs) that were stationed at two locations within Strasbourg, France, during a winter period in 2019/2020. The authors found that the Clemenceau site generally had, on average, higher PM1 than the Danube site due to differences in emissions; however, the composition between the two sites were generally similar (e.g., organics, sulfate, nitrate, ammonium, fossil fuel black carbon, and wood burning biomass burning). Running PMF with the dataset collected from each instrument similarly showed similar organic composition, with the largest difference in the hydrocarbon-like organic aerosol (HOA) at Danube vs Clemenceau. However, if PMF was conducted with the dataset with both instruments as one large dataset, large differences in the organic components occur between Danube than Clemenceau; however, the individual PMF determined for Clemenceau was similar to the combined PMF results for the same location.

As PMF is a tool often used for analysis for investigating sources of both particulate matter and gases, investigating potential sources of uncertainty in this tool is of value for the community of ACP. This paper could potentially also be published in AMT, as it is about the technique of PMF. Either way, the following comments need to be addressed prior to publication.

We thank the reviewer for her/his attention to our manuscript and positive comments.

**Minor**

1) It was not clear until looking at the figure what orifice was used for both ACSMs to know what diameter cut-off the measurements correspond to (e.g., PM1 vs PM2.5).

We specified this now in the main text, section 2.2: "*During winter 2019/2020, the chemical composition of NR-PM$_1$ was investigated using two quadrupole ACSMs (Q-ACSM, Aerosol Chemical Speciation Monitor, Ng et al., (2011)) concomitantly at the Danube and Clemenceau stations. In this instrument, atmospheric particles are sampled at a flow rate of 3 L min-1 (sampling line OD = 9.5 mm; ID = 6.5 mm; 2.2 m long stainless tube) with a cut-off at 2.5 µm*

*using a sampling head, then subsampled at a flow rate of around 85 cc min-1 determined by a 100 μm critical aperture mounted at the instrument inlet* equipped with $PM_1$ aerodynamic lens".

2) How co-located were the AE33, Q-ACSM, and FIDAS 200? E.g., were they sampling from similar inlets for AE33 and Q-ACSM? Were the sampling heights similar for all three instruments? How close were the inlets for all three inlets?

Each instrument has its own distinct sampling line. We have added this information to the main text, section 2.2: "The three measurement instruments, AE33, ACSM, and FIDAS are located in the same station and therefore in exactly the same place. Their sampling lines are separate but only a few meters apart, in accordance with national guidelines, and are set at the same sampling height.".

3) Was there a dryer for any of the instruments?

Yes, we have added this information to the main text as a supplement to comment 2 as well: "The ACSM and AE33 instruments were equipped with a dryer to maintain a relative humidity below 40 %".

4) How statistically different are the average values shown in Table 1? There is discussion about the percent differences in the concentrations; however, the average values fall within the standard deviation, which is assumed to be the spread in the observations and not the uncertainty of the measurements?

We thank the reviewer for this comment. The standard deviation presented in Table 1 reflects the dispersion of observations and not measurement uncertainty. In order to assess the statistical significance of the differences between the average concentrations at the Danube and Clemenceau sites, we performed Welch's t-tests for each species. The tests show that the differences between the average concentrations are statistically significant ($p<0.05$) for all species (see table below).

| Specie | Average Clem. | Average Danube | p_value |
|---|---|---|---|
| OA | 4.38045892 | 3.98188878 | 0.00157018 |
| SO4 | 0.72440656 | 0.59962833 | $3.19E^{-07}$ |
| NO3 | 2.31989462 | 1.83422787 | $1.74E^{-10}$ |
| NH4 | 0.8939168 | 0.73731614 | $7.57E^{-08}$ |
| Cl | 0.04487007 | 0.08217552 | $2.42E^{-33}$ |
| eBCff | 0.83308443 | 0.75991424 | 0.00667625 |
| eBCwb | 0.49663277 | 0.32052488 | $9.07E^{-33}$ |

**Major**

1) Figure S3 does not make sense though it is needed, I believe, for the argument about potentially why the different ACSMs have different PMF results. How is it for both instruments and what is the average mass spectra being compared against? How does it impact total OA?

Figure S3 presents the difference between the mass spectra (normalized to total OA) of both ACSMs during two periods: Upper panel: during the pre-campaign intercomparison exercise in Metz-Borny; Lower panel: during concomitant measurements at both sites in Strasbourg.

2) Figure S6 & Figure 3. Total PM2.5 is generally constant across an urban environment unless there is a very localized emission source, though that emission source maybe more impactful towards PM10 and PM0.1. However, though the PM2.5 (black line) looks generally similar between the two sites in Figure 3, there is very different slopes between the two ACSMs vs PM2.5 in Figure S6 (also, unclear which value is slope vs intercept). What is potentially leading to these differences, and what does it mean for the quantification of one instrument vs another?

As now mentioned in the revised manuscript (thanks for the comment), the a values correspond to the y-intercept and the b values correspond to the slopes (0.62 for ACSM Danube and 0.75 for ACSM Clemenceau). The slopes are not very different for the two sites.

The value of the correlation slope $NR-PM_1 + eBC = f(PM_{2.5})$ provides information on the accuracy of the absolute concentrations measured by the ACSM, whose values, once all technical validation steps have been completed, depend linearly on the ionization efficiency (IE) and the collection efficiency (CE). This value depends on the actual $PM_{2.5}/PM_1$ ratio, which varies according to location and season.

3) The biggest concern and the least amount of discussion is with the combined PMF vs the individual PMF. From the discussion, it is not clear what is the "preferred," more accurate method? E.g., if there are multiple AMSs, ACSMs, or other measurements measuring the composition and concentration of PM, should they be combined into one dataset to conduct PMF for improved accuracy, or was the single PMF more accurate? Was whether one dataset was driving the results of the other dataset investigated? E.g., end points are determined, and then the results are determined from those end points. However, as the authors discuss, one location appeared to potentially have a mixed end-point as they called one of the results COA-like. Does it make sense for the Danube PMF results to have changed so much? I understand that this is a measurement report; however, the results of this paper has large implications for the general understanding and usage of PMF, particularly in how "certain" the results are and how to proceed when there are multiple measurements in one urban location. E.g., are there performance aspects or metrics that should be considered to determine if the PMF may be skewed due to unknown performance of one ACSM, especially if there are not multiple ACSMs to compare against or other external data to compare?

Following the comment #2 from Referee #1, We have added this paragraph in the conclusion part:

"In summary, the comparison of different PMF methods carried out in this study highlights caveats and limitations inherent to such kind of SA approach First, the elucidation of OA sources based on factors derived from PMF should be interpreted with caution considering real-world. In addition, attention should be exercised when combining data from different measurement instruments, as they are not strictly identical in terms of sensitivity. However, positioning two instruments in the same location (or close to each other) can help to verify the presence of atypical or unique factors and explain discrepancies. These limitations introduce uncertainties in the apportionment of OA sources and in the consistency of factor interpretation. In order to improve the identification and interpretation of PMF factors, we propose the integration of complementary datasets (e.g., molecular tracers), which would provide additional constraints. Future work should include a focus on refined methodologies to better handle multi-instrument and multi-timescale datasets, and on the elaboration of standardized protocols for inter-instrument comparisons. Ultimately, improving these methods will lead to a better understanding of the sources, evolution, and role of OA in the atmosphere, which is crucial for accurately assessing their impacts on air quality, health, and climate."

---

## Author Comment (AC2)

**Response to the reviewers on EGUSPHERE-2025-648 "Measurement report: Lessons learned from the comparison and combination of fine carbonaceous aerosol source apportionment at two locations in the city of Strasbourg, France"**

We thank the editor and the reviewers for their constructive advice and comments on our manuscript. In the following, we respond to all reviewers' comments using a black font for original review comments, green font for authors' responses, and blue font for changes in the revised version.

**#Referee 3**

The study by Chebaicheb et. al, provides a standard report of fine aerosol measurements using a combination of ACSM/aethalometer instruments. In terms of the single vs. combined PMF comparison, a solid job has been done, which is more about the measurement technique. In terms of scientific contribution or novelty, I see an interesting discussion on "amine-related OA" factor. With the exception of this, however, the results are fairly standard compared to previously published similar work. For this reason, I recommend even further data exploitation in a situation when two ACSM/AE33 were measured in parallel within the same city. My comments are following.

We thank the reviewer for the positive and constructive comments.

1) Can you present the concentration ratios of the time series between ACSMs to identify which species or m/z are significant for either station even if the distance between sites is not large?

The scatter plots between ACSM#1 (Clemenceau) and ACSM #2 (Danube) for each species (OA, SO4, NO3, NH4, eBCff, and eBCwb), (the b values represent the slopes) are given below and have been added to the supplement (Figure S4). Differences in these time series have been discussed in section 3 of the main text: "The average mass concentrations of NR-PM1 species and eBC presented in Table 1 showed only slight differences between the two sites, with overall higher levels at the Clemenceau site. This could be attributed to the proximity of primary exhaust and non-exhaust emissions from road traffic as well as more intense condensation and coagulation processes. It should also be noted that the environment of the Clemenceau station is more urbanized (city center) compared to the Danube site, which may also partly explain these observations. OA is associated with the highest concentrations at both sites - with values of 4.0 µg m-3 and 4.3 µg m-3 at the Danube and Clemenceau sites, respectively - reinforcing the interest in the apportionment of its main sources. The second main compound at both sites was nitrate, with concentrations about 20% higher at Clemenceau compared to Danube. The differences in sulfate and eBCff concentrations are about 15 % on average (with the highest

concentrations still observed at Clemenceau). Complementarily, results from offline analyses performed on filters collected in February 2020 indicate slightly higher concentrations for Clemenceau (Table 1). Surprisingly, however, filter-based levoglucosan analyses indicate similar concentration levels at both sites while eBCwb appears to be about 40 % higher at Clemenceau, and the comparison of OA mass spectra averaged over the study period also indicates significantly higher signals for the highest m/z's, including common wood-burning tracers (see Figure S3), at Clemenceau.".

[Figure]

Figure S4: Scatter plots between ACSM #1 (further deployed at the Clemenceau site) and #2 (deployed at the Danube site) for the chemical species (OA, NO3, NH4, SO4, eBC$_{ff}$ and eBC$_{wb}$) (the b values represent the slopes).

2) In context of previous comment, the concentration difference in m/z 60 (Fig.S14) looks like an interesting result if we take into account a fairly small distance between these two sites. It probably indicates some close source of fresh emissions from biomass burning. Is this plausible at Strasbourg? When you comparing the instruments against each other (Aug-Oct 2019, Fig.S1), how did the m/z60 comparison come out? If the pre-campaign comparison was the same for m/z60, I would add this figure to the main text and expand discussion on a possible specific source in the vicinity of Clemenceau site.

We compared the m/z60 concentrations during the period (August-October 2019, in Metz) (see figure below), showing higher m/z 60 concentrations with ACSM#1 as discussed in the main text, section 2.2: "As a matter of fact, the few m/z ratios showing the highest concentrations for the under-estimating instrument – which was further installed at the Clemenceau station during the wintertime Strasbourg campaign – included m/z commonly attributed to biomass burning OA (in particular m/z 60 and 73, see Fig. S3)."

[Figure]

3) L. 370, authors state: "A factor profile associated with amine-OA and a specific daily profile are consistent with an industrial source. This factor could therefore be associated with an industrial source of OA." Can you be more specific and maybe even hypothesize about a specific industrial source of this factor? If you know wind direction (Fig.S16) and also that it is a local source, it might not be a problem to pick out something specific. It would be helpful for information if a similar source appears in other papers.

This comment is addressed as follows in the revised version:

"Upon examining the emission inventory, we found a significant amount of particulate emissions linked to an industry zone, which aligns with the pollution roses. Although we found almost no info on the processes used in that type of industry, it seems polyamines are used in the production process of asphalt production as an example (as indicated in the Chinese patent, https://patents.google.com/patent/CN102604125B/en)."

4) In Fig.S4 is a comparison with OC and EC measured on the filters. The agreement is very good. Can you add a similar comparison of filters vs. ACSM concentrations for SO4, NO3 and NH4, and also the discussed comparison of Levoglucosan vs. eBCwb?

We thank the reviewer for this comment. We have added the graphs below comparing the filters and the ACSM/AE33 for the Danube and Clemenceau sites to the Supplement (Figure S6) with this paragraph: "The results for ACSM species (SO$_4$, NO$_3$, and NH$_4$) showed very good correlation coefficient values (r$^2$ > 0.9) for both sites, with ratios of approximately 1 for the Danube site and ratios of approximately 1.2 and 1.3 for the Clemenceau site, showing a good agreement between ACSM chemical species and offline measurements. For the eBCwb vs. levoglucosan comparison, the differences are important with a ratio of 1.5 for the Danube site and around 2.5 for the Clemenceau site. This can be explained by both emission sources and the methodological separation of eBC fractions. As Clemenceau is a traffic-dominated urban site, the separation between eBCwb and eBCff is not always well-defined, leading to potential overestimation of eBCwb and higher eBCwb/levoglucosan ratio.".

**Danube**

**Clemenceau**

[Figure]

Figure S6: Scatter plots of ACSM/AE33 species vs. offline measurements for both Strasbourg sites: Danube and Clemenceau.

5) 3: Can you set the same range on x-axis (time)? It would be better for visual comparing of time series with each other. I also suggest to add Fig.2 on top of Fig.3.

As proposed, we have defined the same range on the x-axis for Figure 3. However, the suggestion to add Figure 2 above Figure 3 cannot be implemented, as Figure 2 is included in section 2.5 and Figure 3 in section 3.

[Figure]

Figure 3: PM$_1$ species at the Danube (top) and Clemenceau (bottom) sites during the studied period.

6) S1 in supplement can be extended by intercomparison of aethalometers prior to the Strasbourg campaigns.

For the campaign conducted in Metz prior to the one in Strasbourg, we did not cross-compare the AE33s, but only the ACSMs.